# SPIRIT and CONSORT extensions for early phase dose-finding clinical trials: the DEFINE (DosE-FIndiNg Extensions) study protocol

Aude Espinasse ![ORCID],[1] Olga Solovyeva,[1] Munyaradzi Dimairo,[2] Christopher Weir ![ORCID],[3] Thomas Jaki,[4,5] Adrian Mander,[6] Andrew Kightley,[7] Jeffry Evans,[8] Shing Lee,[9] Alun Bedding,[10] Sally Hopewell,[11] Khadija Rantell,[12] Rong Liu,[13] An-Wen Chan,[14] Johann De Bono,[15,16] Christina Yap ![ORCID] [1]

AE and OS are joint first authors.

For numbered affiliations see end of article.

**Correspondence to**
Prof Christina Yap;
christina.yap@icr.ac.uk

## ABSTRACT

**Introduction** Early phase dose-finding (EPDF) studies are critical for the development of new treatments, directly influencing whether compounds or interventions can be investigated in further trials to confirm their safety and efficacy. There exists guidance for clinical trial protocols and reporting of completed trials in the Standard Protocol Items: Recommendations for Interventional Trials (SPIRIT) 2013 and CONsolidated Standards Of Reporting Randomised Trials (CONSORT) 2010 statements. However, neither the original statements nor their extensions adequately cover the specific features of EPDF trials. The DEFINE (DosE-FIndiNg Extensions) study aims to enhance transparency, completeness, reproducibility and interpretation of EPDF trial protocols (SPIRIT-DEFINE) and their reports once completed (CONSORT-DEFINE), across all disease areas, building on the original SPIRIT 2013 and CONSORT 2010 statements.

**Methods and analysis** A methodological review of published EPDF trials will be conducted to identify features and deficiencies in reporting and inform the initial generation of the candidate items. The early draft checklists will be enriched through a review of published and grey literature, real-world examples analysis, citation and reference searches and consultation with international experts, including regulators and journal editors. Development of CONSORT-DEFINE commenced in March 2021, followed by SPIRIT-DEFINE from January 2022. A modified Delphi process, involving worldwide, multidisciplinary and cross-sector key stakeholders, will be run to refine the checklists. An international consensus meeting in autumn 2022 will finalise the list of items to be included in both guidance extensions.

**Ethics and dissemination** This project was approved by ICR's Committee for Clinical Research. The Health Research Authority confirmed Research Ethics Approval is not required. The dissemination strategy aims to maximise guideline awareness and uptake, including but not limited to dissemination in stakeholder meetings, conferences, peer-reviewed publications and on the EQUATOR Network and DEFINE study websites.

**Registration details** SPIRIT-DEFINE and CONSORT-DEFINE are registered with the EQUATOR Network.

## STRENGTHS AND LIMITATIONS OF THIS STUDY

⇒ This study will develop international consensus-driven Standard Protocol Items: Recommendations for Interventional Trials (SPIRIT) and CONsolidated Standards Of Reporting Randomised Trials (CONSORT) extensions using a gold standard methodological framework, for early phase dose-finding clinical trials across all disease areas and regardless of trial design used.

⇒ A multidisciplinary international team of experts in both academia and pharmaceutical industry, regulators, SPIRIT and CONSORT group representatives and a patient partner has been brought together to drive the delivery of the project.

⇒ A diverse group of stakeholders, including clinical trial researchers, regulators, ethics committee members, journal editors, funders and funding committee members, and patients and public advocates, will be involved in the Delphi survey and consensus meeting.

⇒ The scope of our guidelines does not specifically cover early phase trials with only one dosing regimen or later phase dose-finding trials with dose (de-)escalations; however, we would expect the basic principles may still be applicable.

⇒ The consensus meeting discussions will not be anonymous, which may impact the flow of dialogue; however, the voting process to determine the inclusion of items will be anonymous.

## INTRODUCTION
### Background
Early phase dose-finding (EPDF) or dose-escalation trials, also referred to as phase I or phase I/II, are critical in clinical therapy development. Depending on the drug and endpoint of interest, the studies may be conducted in healthy volunteers or patients with the condition or disease. These studies involve interim dose decisions and may provide data on safety, adverse effects,

pharmacokinetics (characterisation of a drug's absorption, distribution, metabolism, and excretion), pharmacodynamics, biomarker activity, clinical activity and other information needed to choose a suitable dosage range and/or administration schedule to inform further studies. Results from these trials directly influence decisions on further development and whether the selected doses and schedules are sufficiently safe and have promising results on activity.

A clinical trial protocol is a vital document that details the study rationale, methods, organisation and ethical considerations.[1] By providing the details to guide the conduct of a high-quality study, a well-written protocol is a shared central reference for the study teams[2 3] and facilitates appraisal of its scientific, methodological, safety and ethical rigour by external reviewers. However, protocols can vary greatly in content and quality despite their importance.[2 3] To address this, the Standard Protocol Items: Recommendations for Interventional Trials (SPIRIT) 2013[2] statement was established to provide evidence-based guidance for the minimum essential content of clinical trial protocols and is widely endorsed as the international standard for trial protocols. Although the considerations of SPIRIT 2013 are largely applicable across many types of trials, some circumstances require additional items.[2] Guidance on content specific to EPDF trials, including dose and schedule determination based on safety/tolerability either alone or with one or more pharmacokinetic or activity markers, is lacking. Examples of features unique to such trials include:

► Starting dose and its justification.
► How interim dose decisions will be undertaken (including clearly defined outcome measures and their assessment window, and analysis populations for interim adaptations).
► How future recommended dose(s) will be selected.

Incomplete or unclear information on the design, conduct and analysis in EPDF trial protocols and reporting papers hinders the interpretability and reproducibility of the results from such studies, which may impact the overall clinical development timeline, lead to erroneous conclusions on safety and efficacy, and compromise the safety of trial participants.[4]

This is particularly relevant as a considerable number of early phase trials are sponsored and run by academic institutions or publicly funded organisations with funding from non-commercial sources, including Research Councils and medical charities (eg, Cancer Research UK, Wellcome Trust, US National Cancer Institute). In the UK, 159 out of 1157 (14%) phase I clinical trials, started in 2014–2018, had non-industry sponsors (data from ClinicalTrials.gov). This emphasises the importance of this research to public research institutions and industry alike. Based on results from ClinicalTrials.gov of trials in all countries, there are substantially more phase I trials than phase III trials (13 826 phase I vs 9501 phase III which started in 2014–2018). Data from pharmaceutical trials in the USA in 2004–2012 show that the estimated average cost of a phase I trial across all therapeutic areas ranged from US$1.4 to US$6.6 million[5]; such high costs reinforce the importance of managing resources efficiently. The attrition rate throughout the drug development process is high, and the success rate between phase I studies and marketing authorisation has been reported as between 9.8% and 13.8%,[6 7] with failure being primarily attributable to either poor tolerability or lack of biological activity (79% of failed studies over the period 2016–2018).[8] In this context, EPDF trial results must be assessed accurately to avoid poor dose selection, which will often lead to failed trials (phases II and phase III), delays in regulatory submissions, additional postmarketing commitments or dose changes postapproval due to excessive toxicities or lack of efficacy.[9]

The use of more complex dose-escalation designs such as model-assisted or model-based designs is rising: 1.6% (20/1235 phase I published cancer trials) in 1991–2006[10] to 6.4% (11/172) by 2012–2014.[11] Such designs are more complex to implement[12–14] and require the specification of more design features.[15] Further transparency and reporting demands are needed in such protocols and trial reports to facilitate understanding of the design, ensure the methods and results are reproducible, and explain how dose decisions will be and have been made.[16–18]

More than 580 biomedical journals now require that trial reports conform to the CONsolidated Standards Of Reporting Randomised Trials (CONSORT) 2010 reporting guidelines for randomised parallel group clinical trials or an appropriate CONSORT extension to improve transparency, reproducibility, consistency and accuracy in reporting.[19–21] A total of 153 journals, as well as a growing number of commercial and non-commercial funders, regulators, trial organisations and patient groups, have also endorsed SPIRIT.[22] A systematic review based on more than 16 000 trials published in 2012 showed that journal endorsement of the CONSORT guidelines was associated with more completely reported randomised trials.[23]

Neither the original guidance, SPIRIT 2013 and CONSORT 2010, nor their extensions adequately cover the features of EPDF trials. The DosE-FIndiNg Extensions (DEFINE) study aims to enhance transparency, completeness, reproducibility and interpretation of EPDF trial protocols and their reporting of results, across all disease areas, and to build on the checklists outlined in the SPIRIT 2013 and CONSORT 2010 statements.

## Overall aim

The aim of this research is to develop and disseminate an extension to the SPIRIT 2013 and CONSORT 2010 statements tailored to the specific requirements of EPDF clinical trials across all disease areas.[24] The full study protocols are accessible on the Enhancing the QUAlity and Transparency Of health Research (EQUATOR) website.[25 26]

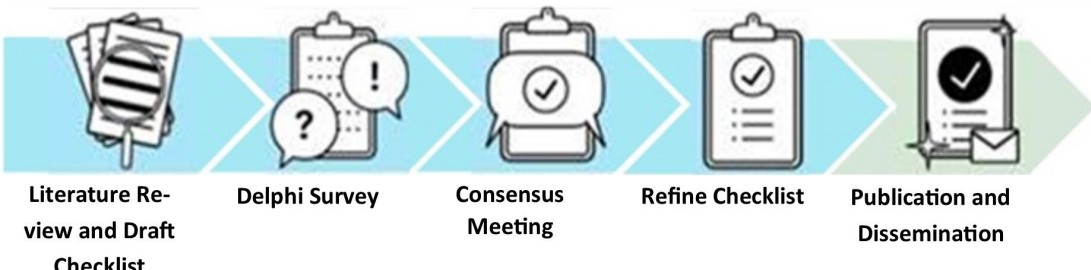

**Figure 1** Project overview for the development of SPIRIT-DEFINE and CONSORT-DEFINE guidelines. CONSORT, CONsolidated Standards Of Reporting Randomised Trials; SPIRIT, Standard Protocol Items: Recommendations for Interventional Trials; DEFINE, DosE-FIndiNg Extensions.

## METHODS AND ANALYSIS

The strategy for the development of reporting guidelines follows the gold-standard methodology framework for guideline development recommended by the EQUATOR network.[27] To ensure the guidance is as impactful and widely adopted as possible, an international Executive Committee was formed, comprising a multidisciplinary team of methodologists, clinicians with expertise in early phase trials in both academia and pharmaceutical industry, a representative each from the SPIRIT and CONSORT groups and a patient and public partner, with planned active engagement with regulators. An independent multidisciplinary expert panel will provide oversight and quality control assurances.

Development of CONSORT-DEFINE commenced in March 2021, followed by SPIRIT-DEFINE from January 2022. Figure 1 illustrates the development process, and each stage will be addressed in detail below.

### Stage 1: literature review and draft checklist generation

The objectives for this stage are to (A) explore current practice in EPDF trials reporting and identify gaps and (B) generate candidate items for CONSORT-DEFINE SPIRIT-DEFINE checklists.

### Methodological review

A methodological review[28] will be conducted to explore the current status of reporting of EPDF trials, identify gaps and specific features of EPDF trials not adequately covered by existing guidance, and inform the drafting of the checklist. The review will also serve in providing a sampling frame for some of the stakeholder categories for the Delphi survey (see the 'Stage 2: Delphi survey section'). A random sample of 476 papers in EPDF trials published between 2011 and 2020, stratified by setting (oncology/non-oncology), will be evaluated. This sample size will provide a two-sided 95% CI for the reporting frequency of an individual item with a width of at most 9% (±4.5%) based on a conservative sample proportion of 0.5 (which gives the largest variance). To standardise the process, a detailed data extraction form and comprehensive guidance will be generated, and agreement between reviewers assessed.

### Candidate item generation

Based on the results of the methodological review as well as expert opinion from the Executive Committee, items considered relevant to constituting a minimum set of reporting requirements will be identified as candidates items for CONSORT-DEFINE. A literature review of multiple databases (Medline via PubMed and Embase) will be performed, alongside grey literature and regulatory or industry guidelines, to identify relevant guidance. Recommendations will also be sought from experts, including regulatory bodies. The SPIRIT-DEFINE candidate item generation process is presented in figure 2 and described below.

An initial draft of the SPIRIT-DEFINE candidate items will be prepared, building on the original SPIRIT 2013, and enriched by the candidate items identified as specific to EPDF trials from the CONSORT-DEFINE development work. The list will be refined through expert opinions from the Executive Committee, grey literature, including regulatory and industry guidance documents and protocol templates by professional groups. Key stakeholder groups identified in the CONSORT-DEFINE development protocol (clinical trials units, including Medicines and Healthcare products Regulatory Agency-accredited phase I units, funders and ethics committees) and experts from other protocol standard initiatives relevant to EPDF trials (eg, from trial registries) will be consulted and their templates included.

Building on the review conducted for CONSORT-DEFINE, the search strategy will be updated to identify protocol recommendations in peer-reviewed literature. Relevant literature not picked up by the search strategy but recommended by experts will be included. Citation and reference searches of key articles will also be conducted. Throughout the stage one (draft checklist generation) process, the Executive Committee will refine the candidate items for both CONSORT-DEFINE and SPIRIT-DEFINE guidance.

### Stage 2: Delphi survey

The draft candidate items for the SPIRIT-DEFINE and CONSORT-DEFINE checklists will be submitted for feedback to a wider stakeholder group through a Delphi survey. The Delphi process will be conducted according

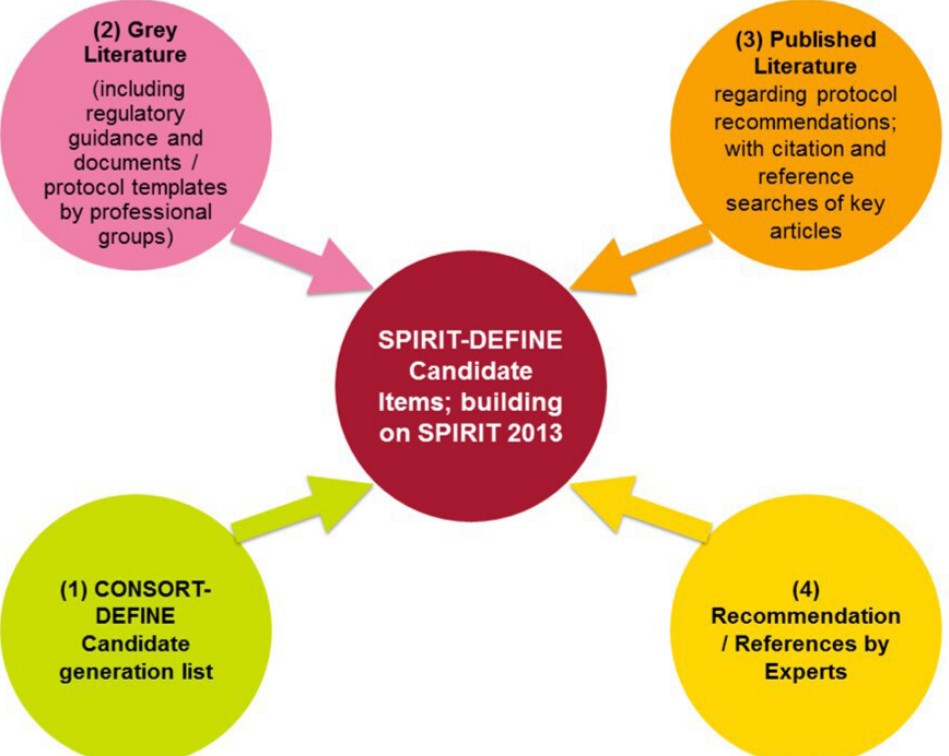

**Figure 2** SPIRIT-DEFINE candidate item generation development process. CONSORT, CONsolidated Standards Of Reporting Randomised Trials; SPIRIT, Standard Protocol Items: Recommendations for Interventional Trials; DosE-FIndiNg Extensions

to existing methodological guidance[29–31] and involves inviting participants to complete iterative rounds of a web-based survey, where results from earlier rounds will inform the design of subsequent rounds. Each candidate item will be scored on a 9-point Likert scale relating to the participant's opinion of its importance, grouped into three categories: (1–3) 'not important', (4–6) 'important but not critical' and (7–9) 'important and critical'. An option 'unable to rate' will be provided for participants who are unable to give their rating opinions for any reason. Free text fields will also be used to elicit comments on the candidate items, and in round 1, participants will also have the opportunity to suggest additional items.

The Executive Committee will discuss the results between each round and agree on any required changes (see the 'Analysis' section). The DEFINE Delphi survey will be hosted on the University of Liverpool's Delphi-Manager, a purpose-built web-based platform and the Executive Committee will pilot the survey before launch.

### Identification of participants

A wide cross-section of stakeholders will be approached to take part in the Delphi survey. For this study, stakeholders will be considered to be direct users or beneficiaries of the guidance and those involved in research conduct, governance, approval, commissioning, funding or publishing EPDF trials.

Potential participants will be approached through a combination of individual and group approaches through publicly available contact details and various professional

organisations or advocacy groups and encouraged to disseminate the invitation further. Professional contacts of the Executive Committee experts will be contacted, and events and conferences used to garner participation. Table 1 references the identified groups as well as contact platforms and organisations. The survey will also be advertised on social media, and a link provided on the DEFINE study website (www.icr.ac.uk/DEFINEstudy).

Consent to take part will be sought via the web-based survey application. No personal identifiable data will be collected aside from name and email address. Data gathered will include professional background characteristics, including geographical location, self-identified stakeholder group (as defined in the 'Identification of participants' section), and years of experience in clinical research and early phase trials. Information on data processing and handling will be provided on the participant information sheet via email invitation and website.

### Sample size

As this is a prospective exercise and a multi-faceted survey, the sample size was decided on pragmatically, to be both achievable and ensure a meaningful representation of all the stakeholder categories. The survey will seek to obtain responses from at least 15 participants in each of the identified stakeholder categories, giving an overall target of at least 90 participants. To achieve this, as many potential participants as possible will be approached, identified through the authors list from the methodological review, approaches from professionals following professional

**Table 1** Delphi survey stakeholders and methods of access

| Stakeholders | Platforms |
|---|---|
| Clinical trials researchers (including clinicians/clinical pharmacologists, trial management staff, statisticians, trial methodologists) | ▶ Medical Research Council-National Institute for Health and Care Research Trial Methodology Research Partnership (MRC-NIHR) (UK)<br>▶ UK Clinical Research Collaboration Network of Registered clinical trial Units<br>▶ Targeted conferences or organisations such as the Society for Clinical Trials, International Clinical Trials Methodology Conference, International Society for Clinical Biostatistics, Statisticians in the Pharmaceutical Industry, European Federation of Statisticians in the Pharmaceutical Industry, Drug Information Association<br>▶ Clinical Conferences such as the National Cancer Research Institute (NCRI) annual conference (NCRI), the European Society for Medical Oncology congress, American Society for Clinical Oncology, the Experimental Cancer Medicine Centres events, European Centre for Rare Diseases and orphan products<br>▶ Sponsors from industry (via organisations such as Pharmaceutical Research and Manufacturers of America in the US, European Federation of Pharmaceutical Industries and Associations in Europe) or the Association of British Pharmaceutical Industry<br>▶ Publications: Corresponding authors of papers selected for the Methodological review as well as papers identified but not sampled. If necessary, further searches without data limitation may be performed<br>▶ Executive Committee members' professional contacts<br>▶ Targeted professional social network groups |
| Regulators | ▶ US Food and Drug Administration<br>▶ European Medicines Agency<br>▶ UK Medicines and Healthcare products Regulatory Agency<br>▶ Japan Pharmaceuticals and Medical Devices Agency<br>▶ China National Medical Product Association Centre for Drug Evaluation.<br>▶ Australia Therapeutic Group Administration<br>▶ Drugs Controller General of India<br>▶ Health Products and Food Branch, Health Canada<br>▶ Ministry of Food and Drug Safety, South Korea<br>▶ Executive Committee members' professional contacts |
| Ethics committee/ethicscommittee members | ▶ UK Health Research Authority (targeting Research Ethics Committees specialised in reviewing early phase trials)<br>▶ European Network of ethics Committees<br>▶ US Institutional Review Boards<br>▶ Australia Health Research Ethics Committees registered through the National Human Medical Research Council.<br>▶ India Institutional Ethics Committees<br>▶ Health Canada and Public Health Agency of Canada Research Ethics Board<br>▶ South Korea Institutes Review Board<br>▶ Executive Committee members' professional contacts |
| Journal editors, associate editors and conference abstracts review committee members | ▶ Leading medical research journals in publishing clinical trials, and targeted journals will be informed by journals where many phase I trials have been published (identified through Methodological review)<br>▶ International Committee of Medical Journal Editors<br>▶ Abstract review Committee members from leading conferences presenting phase one results (see above)<br>▶ Executive Committee members' professional contacts |
| Funders/funding committee members | ▶ Funding panels such as MRC, NIHR, Cancer Research UK, Blood Cancer UK, Wellcome Trust, Bill & Melinda Gates Foundation, Great Ormond Street Hospital and other selected charities funding phase one work as applicable<br>▶ USA National Institutes of Health<br>▶ Pharmaceutical companies<br>▶ Executive Committee members' professional contacts |

Continued

**Table 1** Continued

| Stakeholders | Platforms |
|---|---|
| Patients and public | ▶ Patient and Public engagement platforms<br>▶ European Patients' forum https://www.eu-patient.eu/<br>▶ International disease-specific advocacy groups<br>▶ Patient representatives on phase one trials management groups (through Clinical Trials Units portfolios)<br>▶ Executive Committee members' professional contacts |

meetings and presentations, as well as recommendations from the Executive Committee and independent expert panel. The registration and survey response rates, overall and by stakeholder category and country, will be monitored by the Executive Committee. If a low rate of intake or response is observed, targeted further approaches will be made as appropriate.

## Survey administration

Potential participants will be invited to take part and nominate additional experts to be contacted by the DEFINE team, and various professional or advocacy groups will be approached for dissemination among their members. Interested stakeholders will be asked to register on the survey website before the survey launch. Once registered, consented participants will be alerted to the survey launch by an email containing the link to the survey. Each round of the survey will be open for approximately 4 weeks, and reminders sent weekly during this period. Participants will be allowed to complete a round even if they haven't completed the previous one, provided they have registered for the first round.

## Pilot

The Delphi survey will be piloted by the members of the Executive Committee, before launching the main survey.

Particular attention will be paid to piloting the Delphi survey to ensure patient and public engagement and representation can be optimised. Selected consumer representatives with substantial experience will be approached to take part in the pilot, and their feedback will be sought to ensure the survey is accessible. Should the Delphi survey not allow lay participants to fully contribute due to the complexity, technicality or number of items to be assessed, a focus group will be organised with patient and public involvement and engagement (PPIE) experts to identify a core set of SPIRIT-DEFINE and CONSORT-DEFINE items relevant to PPI contributors. This core set will be submitted for feedback to a wider PPIE audience through a separate process.

## Analysis

The response observed to the initial approaches will be explored in a narrative summary. Following each round, the response rate will be calculated based on the number of participants who registered and completed the survey. A descriptive summary analysis of the responding population will be presented based on the background

characteristics data collected. For each item, the distribution of scores as well as summary statistics (median, IQR, minimum and maximum) will be computed and presented. Summary statistics will be presented by the key stakeholder categories defined in the 'Identification of participants' section and overall. Geographical and professional background characteristics data may be used to explore the data further.

Qualitative data from the free text section of the survey will be thematically analysed to identify potential new items for inclusion.

After each round, members of the Executive Committee will discuss the output and any changes required. Items scored 1–3 'not important' by at least 80% of the participants may be dropped between rounds, subject to confirmation by the Executive Committee. Notes will also be made on any feedback relevant to the development of the explanation and elaboration (E&E) documents.

Participants will also be presented with the distribution of ratings, their ratings from the previous round, as well as feedback on how suggestions and comments from the free text fields were dealt with.

At further rounds, participants will be given the opportunity to change their ratings, and such changes will be monitored. The change in participants' ratings between subsequent rounds will be analysed at item level and interest will be on participants who moved from one category to another (eg, from not 'important' to 'important but not critical').

For each reporting item, the distribution of the changes in rating scores and proportion below 15% change will be reported.

To gauge the level of agreement between round 1 and round 2 ratings, the following statistics will be calculated and reported for each reporting item with associated 95% CIs[32]:
a. Percentage agreement; percentage of participants with the same rating between rounds relative to the total responders to all rounds.
b. Weighted Cohen's kappa coefficient using absolute error weights.[33]

The analysis will be performed in R's latest version at the time of analysis.[34]

## Stopping criteria

The Executive Committee will decide to stop the Delphi survey process once consensus and stability of ratings

have been achieved. It is anticipated that two rounds will be sufficient to achieve this objective; however, the committee may proceed to a third round based on the observed level of agreement and stability and an assessment of whether a subsequent round is likely to yield any further information.

## Stage 3 consensus meeting

The objectives of the consensus meeting will be to finalise the full list of items to be included in the guidance, guided by the information on item importance and level of agreement gleaned during the Delphi survey, as well as the structure of the E&E document. The consensus meeting will follow the recommended methodology for such exercise.[27]

### Definition of consensus

For the purpose of automatic inclusion into the checklist, items rated 7–9 ('critically important') by at least 70% of the Delphi survey respondents will be considered as having reached a consensus.

### Identification of participants

The Executive Committee will be responsible for the selection of relevant experts in each of the key stakeholder categories (see table 1) to be invited to participate in the consensus meeting. Responses to the invitations will be tracked to ensure a balanced representation across the key stakeholder groups.

Checklist items having reached consensus (see the 'Definition of consensus' section) will be automatically recommended for inclusion. Items that did not reach consensus will be discussed for inclusions and/or modification based on the overall importance rating achieved in the last round of the Delphi survey. Following the discussion, consensus group members will anonymously be given an opportunity to make individual decisions about the inclusion of a specific item; 'keep', 'discard', and 'unsure or no opinion'. A decision to retain a reporting item will be based on achieving at least 50% support from group members deciding/wishing to keep the item; however, the Executive Committee will retain the prerogative to discuss and make final decisions for low-scoring items or items where a consensus is difficult to achieve. The rationale to guide decisions will be whether the item addresses elements unique to EPDF trials and whether they belong in a minimum reporting set of items. Notes will be taken, and the discussions audiorecorded, with the participants' consent. Particular attention will be paid to any feedback or discussion requiring inclusion in the E&E document.

Following the meeting, a summary report will be produced and shared with the meeting attendees as well as the Delphi survey participants.

## Stage 4: development of a reporting guidance and explanatory support document

The objectives of this stage are to finalise the SPIRIT-DEFINE and CONSORT-DEFINE guidance and supporting documentation, including the corresponding E&E documents.

After the consensus meeting, the Executive Committee will continue refining the content and wording of both guidelines, and the E&E documents, intended to provide explanations on the rationale and elaboration of the items, as well as evidence and examples applied in the literature. Feedback from the Delphi survey and the consensus meeting will be checked for any information relevant for inclusion in the E&E document.

Both guidelines will be piloted with real-world examples by a selection of key stakeholders with expertise in developing and reporting EPDF trials to test their usability and provide insight into issues that should be addressed in the E&E documents. The committee will discuss feedback from the pilot and decide on further modifications, either to the checklist itself or the E&E document.

## Data management and confidentiality

All data generated and collected during the DEFINE study will be handled, processed and stored according to all applicable data protection legislation. Data collected during the Delphi survey will be stored on a MySQL database hosted on a dedicated DelphiManager server hosted by the University of Liverpool's Data Centre. Following closure of the Delphi survey, data will be downloaded and stored on secure servers at the Institute of Cancer Research Clinical Trials and Statistical Unit, alongside audio recordings and transcripts from the consensus meeting. Access to study data will be restricted to personnel conducting the analyses, and the data will be stored for a minimum of 5 years after the end of the study.

## Patient and public involvement

The DEFINE Study PPIE lead (AK) was involved in the study design from inception and contributed to the development of the protocol. Additional PPIE representatives from both the oncology and non-oncology disease areas will also be consulted on the checklists' items to ensure the optimum representation of this particular patient group. The DEFINE study also comprises a specific PPIE work package aimed at producing lay publications to chart the development of both the SPIRIT-DEFINE and CONSORT-DEFINE guidelines (see the 'Ethics and dissemination' section).

## ETHICS AND DISSEMINATION

This project has been formally assessed for risk and approved by the Institute of Cancer Research Committee for Clinical Research as the sponsor. The Health Research Authority has been consulted and confirmed Research Ethics Approval is not required.

The Executive Committee will devise a detailed dissemination strategy to maximise guideline awareness and uptake. Broadly, the strategy will comprise the following:

► Direct feedback will be provided to the Delphi survey participants, consensus meeting contributors and the stakeholder groups identified in table 1.

- ► The guidelines will be accessible via the CONSORT and EQUATOR network website, as well as on the DEFINE study website, which will also be kept updated throughout the project.
- ► Dissemination at specific UK and international study groups that run phase I trials, such as the UK National Cancer Studies Groups, as well as to funders for early phase trials (including Medical Research Council, Cancer Research UK, National Institute for Health and Care Research (NIHR) Biomedical Research Centres, Experimental Cancer Medicine Centres and National Cancer Institute (NCI)) and industry via The Association of British Pharmaceutical Industry and pharma partners' networks.
- ► Maximising publications in high-impact scientific journals.
- ► Presentation at meetings of UK Clinical Research Collaboration (UKCRC) Clinical Trials Unit, UKCRC Statistics Operational Group and NIHR Early Phase Statistics Group; national and international methodological conferences (eg, International Clinical Trials and Methodology Conference, Society of Clinical Trials or International Society of Clinical Biostatistics) and pharmaceutical conferences/meetings via our industry partners (eg, Statisticians in the Pharmaceutical Industry, European Federation of Statisticians in the Pharmaceutical Industry, Drug Information Association) and clinical conferences (eg, National Cancer Research Institute, European Society for Medical Oncology, American Society for Clinical Oncology, European Centre for Rare Diseases).
- ► Practical Dissemination workshops will be organised, one specifically aimed at journal editors to promote the use of the guideline and encourage endorsement.
- ► Patient and public engagement will be sought via the publication of PPI lay summary papers, including the production of a lay study report template, liaison with patient groups (including the Royal Marsden Patients and Carers Review Panel and the Independent Cancer Patient's Voice), as well as dissemination at local and national PPI events.
- ► Broader communication with the public will also be pursued via the Institute of Cancer Research's website and social media, including blogs, posts on Twitter, Facebook and LinkedIn, press releases and potentially through leadership pieces on trials reporting in the media.

## Author affiliations

[1]Clinical Trial and Statistical Unit, Institute of Cancer Research Sutton, London, UK
[2]School of Health and Related Research, The University of Sheffield, Sheffield, UK
[3]Edinburgh Clinical Trials Unit, University of Edinburgh, Usher Institute, Edinburgh, UK
[4]Computational Statistics Group, Department of Informatics and Data Science, University of Regensburg, Regensburg, Germany
[5]MRC Biostatistics Unit, University of Cambridge, Cambridge, UK
[6]Cardiff University Centre for Trials Research, Cardiff, UK
[7]Patient and Public Involvement Lead, Lichfield, UK
[8]Institute of Cancer Sciences, University of Glasgow, Glasgow, UK
[9]Columbia University, New York, New York, USA
[10]Data and Statistical Sciences Department, Roche Products Ltd, Welwyn Garden City, UK
[11]Centre for Statistics in Medicine, Oxford Clinical Trials Unit, University of Oxford, Oxford, UK
[12]Medicines and Healthcare Products Regulatory Agency, London, UK
[13]Biostatistics Department, Bristol-Myers Squibb Co, New York, New York, USA
[14]Department of Medicine, Women's College Research Institute, University of Toronto, Toronto, Ontario, Canada
[15]Institute of Cancer Research, London, UK
[16]Royal Marsden Hospital, London, UK

**Acknowledgements** The authors acknowledge the support of E. Garrett-Mayer, D. Ashby and J. Isaacs, members of the Independent Expert Panel, in contributing their expertise and support to this initiative, and Siew Wan Hee for proofreading an earlier protocol version. The authors also thank the late D. Altman for his enthusiasm, inspiration and significant contribution to the initial conception of this work.

**Contributors** CY and CW conceived the idea. CY, CW, MD, TJ, AM, AK, JE, SL, SH and JDB obtained funding for CONSORT-DEFINE. AE, OS, MD, CW, TJ, AM, AK, JE, SL, SH, A-WC, JDB and CY contributed to the design of the study. AE, OS and CY wrote the first draft of the manuscript. All authors contributed to the refinement of the study methods and critical revision of the manuscript. All authors read and approved the final version of the manuscript.

**Funding** The CONSORT-DEFINE part of this study was supported by the UK Research and Innovation (UKRI) Medical Research Council-National Institute for Health Research (MRC-NIHR), grant number MR/T044934/1. The SPIRIT-DEFINE part received no specific grant from any funding agency in the public, commercial or not-for-profit sectors. T Jaki received funding from UK Medical Research Council (MC_UU_00002/14). CW was supported in this work by NHS Lothian via Edinburgh Clinical Trials Unit. SL was funded through the National Center for Advancing Translational Sciences, National Institutes of Health (Grant Number UL1TR001873). However, research outputs will be published in line with the funders' publication policy requirements. SPIRIT-DEFINE part did not receive any external funding. For the purpose of open access, the author has applied a Creative Commons Attribution (CC BY) licence to any Author Accepted Manuscript version arising from this submission.

**Disclaimer** The funders have no involvement in the study design, collection, analysis, interpretation of findings, and reporting. The views expressed in this article are those of the author(s) and not necessarily those of the NHS, the NIHR, or the Department of Health.

**Competing interests** JDB has served on advisory boards and received fees from many companies, including Amgen, Astra Zeneca, Astellas, Bayer, Bioxcel Therapeutics, Boehringer Ingelheim, Cellcentric, Daiichi, Eisai, Genentech/Roche, Genmab, GSK, Harpoon, ImCheck Therapeutics, Janssen, Merck Serono, Merck Sharp & Dohme, Menarini/Silicon Biosystems, Orion, Pfizer, Qiagen, Sanofi Aventis, Sierra Oncology, Taiho, Terumo, Vertex Pharmaceuticals. JDB is an employee of The Institute of Cancer Research, which has received funding or other support for his research work from AZ, Astellas, Bayer, Cellcentric, Daiichi, Genentech, Genmab, GSK, Janssen, Merck Serono, MSD, Menarini/Silicon Biosystems, Orion, Sanofi Aventis, Sierra Oncology, Taiho, Pfizer, Vertex, and which has a commercial interest in abiraterone, PARP inhibition in DNA repair defective cancers and PI3K/AKT pathway inhibitors (no personal income). JDB was named as an inventor with no financial interest in Patent No. 8822438 submitted by Janssen that covers the use of abiraterone acetate with corticosteroids. He has been the CI/PI of many industry-sponsored clinical trials. JDB is a National Institute for Health Research (NIHR) Senior Investigator. JE has received advisory board or speaker fees from AstraZeneca, Bayer, Bristol-Myers Squibb, Bicycle Therapeutics, Celgene, Clovis, Eisai, Medivir, Nucana, and Roche/Genentech; and has received funding or other support for non-commercial and commercial studies from Adaptimmune, AstraZeneca, Astellas, Basilea, Bayer, Boehringer Ingelheim, Bicycle Therapeutics, Bristol-Myers Squibb, Beigene, Celgene, Codiak, CytomX, Eisai, GlaxoSmithKline, Immunocore, iOncture, Johnson and Johnson, Lilly, Medivir, Merck Sharp & Dohme, MiNa Therapeutics, Novartis, Nucana, Pfizer, and Roche/Genentech, Sanofi, Sapience Therapeutics, Seagen, Sierra, Starpharma, UCB and Verastem. Professor Evans serves as a member of the Clinical Experts Review Panel and Clinical Research Committee for Cancer Research UK, a member of the International Liver Cancer Association Annual Meeting abstracts committee and a member of Pancreatic Cancer Research Fund Scientific Advisory Panel. JE is also a member

of the American Association for Cancer Research, the American Society of Clinical Oncology, the Association of Cancer Physicians (UK), the British Association for Cancer Research, the European Association for Cancer Research, and the International Liver Cancer Association. JE is a Clinical Subject editor for the British Journal of Cancer and has received honorarium payable to the employing institution for serving as chair of the Independent Data Monitoring Committee for a phase I trial.

**Patient and public involvement**  Patients and/or the public were involved in the design, or conduct, or reporting, or dissemination plans of this research. Refer to the Methods section for further details.

**Patient consent for publication**  Not applicable.

**Provenance and peer review**  Not commissioned; externally peer reviewed.

**ORCID iDs**
Aude Espinasse http://orcid.org/0000-0002-1271-302X
Christopher Weir http://orcid.org/0000-0002-6494-4903
Christina Yap http://orcid.org/0000-0002-6715-2514

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
