## [Reviewer comments · BMJ Open]

ARTICLE DETAILS

TITLE (PROVISIONAL)	SPIRIT and CONSORT extensions for early phase dose-finding clinical trials: the DEFINE (DosE FIndiNg Extensions) study protocol
AUTHORS	Espinasse, Aude; Solovyeva, Olga; Dimairo, Munyaradzi; Weir, Christopher; Jaki, Thomas; Mander, Adrian; Kightley, Andrew; Evans, Jeffrey; Lee, Shing; Bedding, Alun; Hopewell, Sally; Rantell, Khadija; Liu, Rong; Chan, An-Wen; De Bono, Johann; Yap, Christina

VERSION 1 – REVIEW

REVIEWER	Yuan, Ying MD Anderson Cancer Center
REVIEW RETURNED	26-Nov-2022

GENERAL COMMENTS	This manuscript is very well written. The DEFINE study is well thought out with clear and concrete steps. The proposed procedures are reasonable and practical. The only comment I have is that some discussion on the fallback plan may be useful. For example, if one step fails to achieve its target, how to proceed to the next step. There are a few typos or format issues: 1. line 71, "The stakeholders we will..."2. line 173, the title, the first letter of each word should be capitalized or in lower case.3. line 372, "stored on stored in a MySQL..."
--

REVIEWER	Oron, Assaf Institute for Health Metrics and Evaluation, University of Washington
REVIEW RETURNED	12-Jan-2023

GENERAL COMMENTS	Thank you for approaching me to review this article, it is an honor. This is a highly worthy effort, as early phase / dose-finding trials (EPDFs) are far more of a "Wild West" than latter-phase trials, actually more than most human-subject study types. Therefore, more standardization of planning and reporting could be enormously helpful if done right. And you are certainly approaching it in a careful and inclusive manner. *** General/major comments 1. I have not heard of this effort before, so this is related to my first request or recommendation. Despite there being far more EPDFs than late-phase trials, EPDF design/analysis experts are fewer and
---

farther between. While design experts is only a small subset of the overall set of communities you plan to approach, it might be the most difficult to reach out to. I urge you to consider, besides the professional association route, a complementary way to locate such experts who might be lone beasts or retired/etc., based on a targeted search of methodological articles, and adding to your list of contacts authors with a certain combination of publications and citations on the topic in the past ~20 years or so.

2. I wonder how you plan to define EPDFs. If this is only for official Phase I, it might exclude entire bodies of work that would benefit from this immensely. For example, dose-finding studies are extremely common in anesthesiology, but usually they are "post-Phase-III" because they play with the dosage of approved agents. Or they could be "Phase 0", playing with dosage of agents and combinations that were approved separately, but in combination are essentially off-label despite being under routine use. Including such studies means later on, requiring them to come clean regarding their long-term goal. Are you just trying to publish a paper? Or to actually affect broader practice? If the latter, then it's probably a clinical trial and not just a dose-finding exercise.

So: is EPDF "early phase AND dose-finding" or - "early phase OR dose-finding?" I strongly recommend the latter. Just like in CONSORT and SPIRIT guidelines there are optional sections relevant only for subtypes of trials, your guidelines can do the same and maintain a Big Tent.

This also pertains to your 4th bullet point under "strengths and limitations" (p. 3, line 75), in which you exclude the "early-phase only"; i.e., you exclude single-treatment early-phase trials.

But single-treatment early-phase trial might still benefit from additional rigor and specific structure. In fact, some of the most horrific recent trial misdeeds have occurred in precisely these trials, which all too often go under the radar (e.g., TGN1412). While dose-escalation aspects would not be relevant to them, these trials would greatly benefit from a far more detailed checklist to go through regarding benefit vs. risk, how to manage the risk, what are the actual primary/secondary goals of the trial, etc.

It's unfortunately common in Phase I, particularly in cutting-edge interventions where such scrutiny is most needed, to write "safety" in the proper protocol entry for primary endpoint, but then actually conduct the trial and focus the attention as if efficacy/activity output is primary, while keeping safety somewhat on the back burner. This can happen in both dose-finding and single-treatment contexts, but I fear that the latter are even less scrupulous.

3. Your article mentions initial stages completed March 2021 and January 2022. Are there any interim results from that, or is this manuscript the sole share-able output thus far?

*** Specific/minor comments

1. p. 6 line 131 "The use of more efficient but undoubtedly more complex designs:" while being "more efficient" has been the sales slogan for many novel complex designs, this has not been demonstrated in unequivocal fashion. The extremely simple family of up-and-down dose-finding designs gives the most sophisticated

	designs a run for their money in terms of efficiency. See e.g., Oron and Hoff, Clinical Trials 10 (2013), 63-92. So I would revise/remove this efficiency claim, and focus on the complexity and the often associated loss of transparency. 2. p. 6 lines 313-315, perhaps add Maxwell's correlation coefficient (a.k.a. Phi) to the measures evaluating agreement. Last I checked it seemed to be held in higher esteem than kappa. But having both would be nice. 3. p. 13 line 316, why not switch to "R, latest stable version at time of analysis", and obviate the need to remember to change the version at every edit? GOODLUCK!!! Assaf
--	---

VERSION 1 – AUTHOR RESPONSE

Reviewer: 1

Dr. Ying Yuan, MD Anderson Cancer Center Comments to the Author:

This manuscript is very well written. The DEFINE study is well thought out with clear and concrete steps. The proposed procedures are reasonable and practical. The only comment I have is that some discussion on the fallback plan may be useful. For example, if one step fails to achieve its target, how to proceed to the next step.

Thank you for this helpful suggestion Our strategy for generating awareness of the project and engagement to the Delphi and Consensus meeting stages was both through formal approaches as evidenced in Table 1 as well as informal reaches through professional contacts and networks from all members of the Executive Committee. The strategy included monitoring the registration rates both overall and by geographical area and stakeholder category. This allowed us to purposefully include additional potential participants. and target certain categories such as journal editors. As a result, over 1600 potential participants were approached, allowing the project to progress smoothly through the different phases. We have tried to highlight this better in the revision, please see section 2 "Sample size. .

There are a few typos or format issues:

1. line 71, "The stakeholders we will..."
2. line 173, the title, the first letter of each word should be capitalized or in lower case.
3. line 372, "stored on stored in a MySQL..."

Thank you. All typos above have been corrected.

Reviewer: 2

Dr. Assaf Oron, Institute for Health Metrics and Evaluation Comments to the Author:

Thank you for approaching me to review this article, it is an honor. This is a highly worthy effort, as early phase / dose-finding trials (EPDFs) are far more of a "Wild West" than latter-phase trials, actually more than most human-subject study types. Therefore, more standardization of planning and reporting could be enormously helpful if done right. And you are certainly approaching it in a careful and inclusive manner.

*** General/major comments

1. I have not heard of this effort before, so this is related to my first request or recommendation. Despite there being far more EPDFs than late-phase trials, EPDF design/analysis experts are fewer and farther between. While design experts is only a small subset of the overall set of communities you plan to approach, it might be the most difficult to reach out to. I urge you to consider, besides the professional association route, a complementary way to locate such experts who might be lone beasts or retired/etc., based on a targeted search of methodological articles, and adding to your list of contacts authors with a certain combination of publications and citations on the topic in the past ~20 years or so.

Thank you for this helpful suggestion. We will consider this in our dissemination plan of the final guidance. We agree with the reviewer that EPDF experts are fewer than later phase trials. Hence, we have contacted corresponding authors from over 500 early phase I/II dose-finding trial articles in a broad range of trial designs and therapeutic settings published in 2011 to 2020 within our methodological review to take part in the various stages of the project (Delphi Survey/Consensus Meeting as appropriate). Authors from papers identified through the search but not included in the review were also contacted, and a further search conducted without any limitation on dates to further identify experts in geographical areas that were less represented. Our dissemination plan also included promoting the project at several early phases events to elicit interest. Additionally, the Executive Committee and Independent Expert Committee, formed of experts in the field has also been consulted for suggestions from their professional contacts. We have clarified this in section 1 "Identification of participants", in table 1 as well as section 2 "sample size of the revision.

2. I wonder how you plan to define EPDFs. If this is only for official Phase I, it might exclude entire bodies of work that would benefit from this immensely. For example, dose-finding studies are extremely common in anesthesiology, but usually they are "post-Phase-III" because they play with the dosage of approved agents. Or they could be "Phase 0", playing with dosage of agents and combinations that were approved separately, but in combination are essentially off-label despite being under routine use. Including such studies means later on, requiring them to come clean regarding their long-term goal. Are you just trying to publish a paper? Or to actually affect broader practice? If the latter, then it's probably a clinical trial and not just a dose-finding exercise.

So: is EPDF "early phase AND dose-finding" or - "early phase OR dose-finding?" I strongly recommend the latter. Just like in CONSORT and SPIRIT guidelines there are optional sections relevant only for subtypes of trials, your guidelines can do the same and maintain a Big Tent.

This also pertains to your 4th bullet point under "strengths and limitations" (p. 3, line 75), in which you exclude the "early-phase only"; i.e., you exclude single-treatment early-phase trials.

But single-treatment early-phase trial might still benefit from additional rigor and specific structure. In fact, some of the most horrific recent trial misdeeds have occurred in precisely these trials, which all too often go under the radar (e.g., TGN1412). While dose-escalation aspects would not be relevant to them, these trials would greatly benefit from a far more detailed checklist to go through regarding benefit vs. risk, how to manage the risk, what are the actual primary/secondary goals of the trial, etc.

It's unfortunately common in Phase I, particularly in cutting-edge interventions where such scrutiny is most needed, to write "safety" in the proper protocol entry for primary endpoint, but then actually conduct the trial and focus the attention as if efficacy/activity output is primary, while keeping safety somewhat on the back burner. This can happen in both dose-finding and single-treatment contexts, but I fear that the latter are even less scrupulous.

Thank you for this important point. At the inception of the project, the Executive Committee had extensive discussions to define the scope of the guidelines. The guidelines do primarily focus on early phase clinical trials (typically referred to as Phase I with or without dose expansion cohorts or Phase I/II), where interim dose-decisions are taken using accumulating trial data to either escalate, de-escalate, stay at the current level or stop a trial early. The dose assignment decisions could be based on safety, pharmacokinetic, pharmacodynamic or biological markers or a combination of these parameters. It was felt by the group these trials share common specific features not adequately addressed by CONSORT 2010 and SPIRIT 2013 nor any extensions, and that the choice of the easily identified scope explained above would increase the usefulness of the guidelines and affect practice. We agree with the reviewer that scrutiny and additional rigor and structure could indeed benefit a much wider body of works in the early phase single arm setting and potentially later phase post licensing dose finding arenas. Indeed, some aspects of the guidelines will be applicable beyond the boundaries of the trials defined in our scope, and we have now highlighted the relevance and applicability of the guidelines in the section on "Strengths and Limitations" and will also do so our subsequent guideline papers.

3. Your article mentions initial stages completed March 2021 and January 2022. Are there any interim results from that, or is this manuscript the sole share-able output thus far?

We believe the reviewer meant "commenced" rather than "completed". CONSORT-DEFINE and SPIRIT-DEFINE commenced in March 2021 and January 2022 respectively. As this is a protocol paper for the DEFINE study, no interim results are included. We have however presented the initial results of this work at several conferences. Several publications based on the results of the development process and the final guidance are planned.

*** Specific/minor comments

1. p. 6 line 131 "The use of more efficient but undoubtedly more complex designs:" while being "more efficient" has been the sales slogan for many novel complex designs, this has not been demonstrated in unequivocal fashion. The extremely simple family of up-and-down dose-finding designs gives the most sophisticated designs a run for their money in terms of efficiency. See e.g., Oron and Hoff, *Clinical Trials* 10 (2013), 63-92. So I would revise/remove this efficiency claim, and focus on the complexity and the often associated loss of transparency. Many thanks for highlighting this. This has been amended.

2. p. 6 lines 313-315, perhaps add Maxwell's correlation coefficient (a.k.a. Phi) to the measures evaluating agreement. Last I checked it seemed to be held in higher esteem than kappa. But having both would be nice.

Thank you for your suggestion. Our protocol pre-specifies using weighted Cohen's kappa coefficient and absolute error weights as the primary method of evaluating agreement. Further work may be done to explore results of the Delphi Survey in more depth, using alternative statistical methods.

3. p. 13 line 316, why not switch to "R, latest stable version at time of analysis", and obviate the need to remember to change the version at every edit?

Thank you. This has been amended.

We trust these responses will be satisfactory, and remain at your disposal for any further information.

Yours Sincerely,

Christina Yap PhD CStat (on behalf of the co-authors)
Professor of Clinical Trials Biostatistics

VERSION 2 – REVIEW

REVIEWER	Yuan, Ying MD Anderson Cancer Center
REVIEW RETURNED	23-Feb-2023

GENERAL COMMENTS	The authors have addressed my comments. I recommend accept for publication.
---

REVIEWER	Oron, Assaf Institute for Health Metrics and Evaluation, University of Washington
REVIEW RETURNED	11-Mar-2023

GENERAL COMMENTS	Thank you for your kind and detailed responses. I have no further comments. Good luck! Assaf
--